# Durable CD8 T Cell Memory against SARS-CoV-2 by Prime/Boost and Multi-Dose Vaccination: Considerations on Inter-Dose Time Intervals

**DOI:** 10.3390/ijms232214367

**Published:** 2022-11-19

**Authors:** Ambra Natalini, Sonia Simonetti, Carmel Sher, Ugo D’Oro, Adrian C. Hayday, Francesca Di Rosa

**Affiliations:** 1Institute of Molecular Biology and Pathology, National Research Council of Italy (CNR), 00161 Rome, Italy; 2Immunosurveillance Laboratory, The Francis Crick Institute, London NW1 1AT, UK; 3Medical Oncology Department, Campus Bio-Medico University, 00128 Rome, Italy; 4Sackler Faculty of Medicine, Tel Aviv University, Tel Aviv 69978, Israel; 5GSK, 53100 Siena, Italy; 6Peter Gorer Department of Immunobiology, King’s College London, London WC2R 2LS, UK; 7National Institute for Health and Research (NIHR) Biomedical Research Center (BRC), Guy’s and St Thomas’ NHS Foundation Trust, King’s College London, London WC2R 2LS, UK

**Keywords:** immunological memory, vaccination, COVID-19, CD8 T cells

## Abstract

Facing the COVID-19 pandemic, anti-SARS-CoV-2 vaccines were developed at unprecedented pace, productively exploiting contemporary fundamental research and prior art. Large-scale use of anti-SARS-CoV-2 vaccines has greatly limited severe morbidity and mortality. Protection has been correlated with high serum titres of neutralizing antibodies capable of blocking the interaction between the viral surface protein spike and the host SARS-CoV-2 receptor, ACE-2. Yet, vaccine-induced protection subsides over time, and breakthrough infections are commonly observed, mostly reflecting the decay of neutralizing antibodies and the emergence of variant viruses with mutant spike proteins. Memory CD8 T cells are a potent weapon against viruses, as they are against tumour cells. Anti-SARS-CoV-2 memory CD8 T cells are induced by either natural infection or vaccination and can be potentially exploited against spike-mutated viruses. We offer here an overview of current research about the induction of anti-SARS-CoV-2 memory CD8 T cells by vaccination, in the context of prior knowledge on vaccines and on fundamental mechanisms of immunological memory. We focus particularly on how vaccination by two doses (prime/boost) or more (boosters) promotes differentiation of memory CD8 T cells, and on how the time-length of inter-dose intervals may influence the magnitude and persistence of CD8 T cell memory.

## 1. Introduction to COVID-19 and CD8 T Cells

Shortly after an alarming outbreak of a pneumonitis of unknown origin in China in December 2019, thorough investigations allowed rapid discovery of the causative pathogen and determination of its genome sequence, creating the basis for prompt vaccine development. The pathogen was a new coronavirus, which was named SARS-CoV-2, while its corresponding infectious disease was termed COVID-19. It was soon found that most SARS-CoV-2 infected individuals experience few if any symptoms, and that in those who do experience symptoms, COVID-19 is a highly heterogeneous disease. Most COVID-19 patients exhibit respiratory symptoms, ranging from a mild to a severe pneumonia. Other clinical manifestations include gastrointestinal, renal and cardiovascular symptoms. Death can occur due to serious complications including acute respiratory distress syndrome, septic shock, severe coagulopathy and/or multi-organ failure. In March 2020, COVID-19 cases were so numerous and extensively distributed across the globe that the World Health Organization (WHO, Geneva, Switzerland) declared a COVID-19 pandemic.

In a large part, the global response was societal lockdown, with massive implications for the ongoing treatment of chronic diseases, including cancer. Thus, there was a possibly unprecedented drive towards the development of vaccines against SARS-CoV-2 and based on decades of existing and ongoing experimental and theoretical work on immunology in general, and on innovative vaccine platforms in particular, mRNA- and recombinant adenoviral vector-based vaccines were produced and approved with striking rapidity [1,2,3,4]. Their widespread roll-out began between the end of 2020 and the beginning of 2021. Subsequently, many other platforms have been exploited for the development of very many anti-SARS-CoV-2 vaccines, including those based on DNA, proteins and adjuvant, and inactivated virus, etc. [4]. Thus, about two and a half years after the first cases of COVID-19, roughly a dozen anti-SARS-CoV-2 vaccines are currently in use, and many more are at different stages of development. 

Although some of the proposed anti-SARS-CoV-2 vaccines were abandoned after testing in trials, most have proven immunogenic and efficacious in greatly limiting the severity of COVID-19 [5,6,7,8]. Nevertheless, vaccine-induced memory tends to wane over time and SARS-CoV-2 infection breakthroughs are increasingly common in vaccinated persons, albeit with only mild symptoms in most of the cases [9,10]. Currently, a number of major challenges exist, including the emergence of SARS-CoV-2 variants of concerns (VOCs) possibly requiring ad hoc tailored vaccines, difficulties in large-scale vaccine production, and unequal distribution of available vaccines across the globe [11,12]. These challenges need to be met, given that anti-SARS-CoV-2 vaccines undoubtedly remain the most effective defence against COVID-19.

In fact, not only have anti-SARS-CoV-2 vaccines had overall success as public health measures, but they have also provided us with an unprecedented opportunity to deepen our understanding of the mechanisms underlying the human immune response to infections and vaccinations. Already in 2020, scientists equipped with state-of-the-art immuno-monitoring protocols, honed over the past decade, had for the first time the possibility to study blood samples from hundreds of persons contemporaneously infected with the same pathogen. Soon after that, the opportunity arose to investigate, on an unprecedentedly large-scale, the responses to vaccination of naïve individuals of different ages, either in a healthy state or with underlying disease conditions, as well as of individuals previously exposed to SARS-CoV-2.

In providing considerable amounts of data, as evidenced by many thousands of publications, those studies have pinpointed many gaps that remain to be filled, particularly in relation to how immunological memory, a key feature of adaptive immunity, is maintained by B cells and antibodies, and by CD4 and CD8 T cells. In the case of CD8 T cells, our knowledge is insufficient for a variety of reasons, e.g., technical difficulties in measuring the cells’ fine specificities, particularly given the extensive HLA-I polymorphism; lack of correlates of CD8 T cell-mediated protection; challenges in viral epitope profiling, etc. [13]. Yet, CD8 T cell cytotoxicity plays a crucial role against viral infections, as it does against tumours, highlighting the importance of using COVID-19 studies as a springboard for gaining more knowledge of the mechanisms underlying durable vaccine-induced CD8 T cell responses.

Here, we shall offer an overview of the current understanding of the immunological mechanisms underlying CD8 T cell memory induced by vaccination against SARS-CoV-2. Immunological insights provided by the recent use of SARS-CoV-2 vaccines will be discussed in the context of previous data from the longstanding employment of other vaccines, focusing primarily on fundamental mechanisms and pointing out unresolved questions in the field. Special attention will be given to the reasons behind the need to administer two or more vaccine doses to fully establish a durable CD8 T cell memory, and to the mechanisms underlying the impact of the inter-dose time interval on memory CD8 T cell differentiation.

We apologize to the many authors of COVID-19 vaccine studies who we may have failed to cite owing to our particular perspective and/or to space limits. In lieu of this, we refer the readers to a more comprehensive overview provided by excellent reviews such as those by Sette and Crotty and by Goldblatt and colleagues (e.g., [14,15]).

## 2. Vaccine-Induced CD8 T Cell Responses: A General Overview

Induction of an immune response to a vaccine relies on cooperation between the innate and adaptive immunity, as response is started by productive interactions between professional antigen presenting cells (APCs) and naïve T cells. Interestingly, vaccines are commonly injected intramuscularly, although skeletal muscles are anatomical districts where local APC populations are relatively poorly understood. It is commonplace to consider that at the site of vaccine injection, dendritic cells (DCs), i.e., professional APCs which belong to the innate immune system, can engulf vaccine components, including vaccine-carried antigenic proteins, and, after intracellular processing, expose on their surface vaccine-derived antigenic peptides within the pocket of either MHC-I or MHC-II molecules (MHC is known as HLA in humans). Such DCs migrate to tissue-draining lymph nodes (LNs), wherein they present antigenic peptides to recirculating naïve T cells. Notably, activated or mature DCs present to T cells antigen–MHC complexes (signal 1) and ligands for costimulatory receptors (signal 2) that are necessary to prime naïve T cells towards immunogenic rather than tolerogenic responses. Cytokines secreted by mature DCs and other cells in the tissue (signal 3) further regulate T cell priming, particularly influencing the complexion of the T cells’ effector programming [16,17].

In the case of mRNA-based and DNA-based vaccines, and adenoviral vectors, the antigen-encoding genetic material is taken up and expressed by myocytes at the intramuscular injection site, resulting in direct MHC-I restricted antigen presentation by these cells [18,19]. Nevertheless, priming of naïve T cells, which is particularly relevant to novel infectious agents such as SARS-CoV-2, requires antigen presentation by specialized bone marrow (BM)-derived cells, particularly mature DCs [18,19,20]. DCs have the capacity to take up antigen-containing cell debris at the site of vaccine injection, and present antigenic peptides in MHC-I and MHC-II to naïve CD8 and CD4 T cells, respectively. When DCs are activated or mature, these cognate cellular interactions result in naïve T cell priming.

DC activation or maturation consists of a shift from a resting to a mature state, characterized by remarkable changes in the expression of chemokine receptors, costimulatory molecules, cytokines, etc., that collectively enable mature DCs to prime naïve T cells [21,22]. In healthy tissues, DCs are normally diffusively distributed in a resting or immature state. Upon vaccine injection, a local innate response is initiated by pathogen associated molecular patterns (PAMPs) carried by the vaccine, and/or damage associated molecular patterns (DAMPs) released from host cells that are stressed or injured by vaccination [23,24,25]. Both PAMPs and DAMPs are able to signal through innate immunity receptors expressed by DCs and other host cells, triggering an inflammatory response that results in DC maturation [26]. In the case of vaccines made of attenuated pathogens, e.g., yellow fever vaccine, strong innate and adaptive responses are triggered by the integration of signals derived from PAMPs, DAMPs, and a myriad immunodominant and subdominant antigens, etc., as the vaccine de facto mimics the corresponding natural infection. Thus, this type of vaccine formulation would seem to be optimal for efficacy, but predictably carries risks for immuno-compromised subjects, who belong to a high-priority group to be protected in the event of a pandemic.

In contrast, vaccines not composed of attenuated pathogens typically contain an element called an “adjuvant” capable of triggering innate immunity, and more specifically DC maturation and inflammatory cytokine release. A good example is Freund’s adjuvant, often used in experimental models, which contains microbe-derived molecules that are recognized as PAMPs by innate immunity receptors. The need for this vaccine component was recognized over 30 years ago by Janeway, who coined the term “Immunologist’s dirty little secret” to highlight the emerging realization that antigen alone is insufficient for effective immunization [23]. Some vaccines that are used against COVID-19, e.g., those based on replication defective adenoviruses, carry antigen and adjuvant in the same vector, without the need for additional components [18].

Given the immense diversity of the TCR repertoire of naïve T cells, it is evident that only very small numbers of CD4 and CD8 T cells would be specific for SARS-CoV-2 vaccine-derived antigens [27]. After a lag of 3–5 days following primary immunization, a time that is necessary to enable productive naïve T cell–APC encounters, the few “cognate” antigen-reactive T cells expand exponentially and differentiate, generating a progeny of cells usually detectable by day 7 post-prime. This progeny includes short-lived cells able to exert immediate effector functions upon antigen–MHC triggering, and cells with durable survival and stem-like properties that collectively compose immunological memory pools. Those cells are themselves heterogeneous, displaying different phenotypes and migratory pathways that combine to define central memory (T_CM_), effector memory (T_EM_), recirculating, and tissue-resident memory T cells (T_RM_). These subsets are largely distinguished from each other based on homing receptor expression, localization in the body, and transcriptional and epigenetic regulators that will determine the cells’ survival in their respective niches and their responsiveness to activation [28].

Help for B cell responses, cytokine release, and cytotoxicity are the main T cell effector functions that support anti-pathogen protection. Furthermore, CD4 T cells provide help for naïve CD8 T cell responses, which occur via “licensing” of antigen-presenting DCs [29,30,31,32]. In fact, naïve CD8 T cell priming relies on a two-step process: firstly, a DC presents MHC-II–antigen to a helper CD4 T cell, which after activation provides a CD40L-mediated signal back to the DC, enabling a change in the costimulatory capacity of this cell (DC “licensing”); secondly, the “licensed” DC presents MHC-I–antigen to a naïve CD8 T cell [29,30,31,32].

At the end of the acute phase of the primary response, most effector T cells die in a so-called contraction phase, leaving behind only a small number of long-lived memory T cells that are able to display a more effective and rapid protective response than that provided by naïve T cells, in case of subsequent encounter with the pathogen. For many years various data sets have been used to argue that memory T cells emerge primarily by further differentiation of cells within an effector pool [33,34], or conversely, that they may arise by a separate pathway of differentiation not contingent on passage through an effector stage [35,36]. This topic is still highly debated. It should be noted that during infection, the highest immediate priority is the development of effectors, whereas for prophylactic vaccination, it is important to generate memory cells capable of prompt responses to protect against future infection. Hence, the fundamental importance in vaccine design of understanding how to exploit cues that shift towards memory T cell generation, and, within that, the appropriate balance of memory T cell subtypes. Likewise, vaccine design also needs to accommodate studies over the past decades that have shown that CD8 T cells can enter a dysfunctional or exhausted differentiation state, resulting from inappropriate overstimulation, which may require re-setting by treatment with checkpoint inhibitors [37,38].

## 3. Anti-SARS-CoV-2 CD8 T Cells: Natural Infection versus Vaccination

Emerging evidence suggests that antigen-specific CD8 T cells display a protective function from COVID-19 acute disease caused by natural SARS-CoV-2 infection, as predicted based on the known capacity of these cells to recognize a large repertoire of viral epitopes on the surface of virus-infected cells (Figure 1A). An insightful study in a cohort of critically ill patients showed a correlation between survival and high levels of polyfunctional, PD-L1^+^CXCR3^+^ effector CD8 T cells and of CD8 T cells specific for SARS-CoV-2 nucleocapsid in the blood, suggesting that CD8 T cells had managed to limit naturally established SARS-CoV-2 infection in patients with better prognosis [39]. Moreover, CD8 T cell localization at one of the target tissues of infection was demonstrated in another study that, by examining nasal mucosa, showed local persistence of SARS-CoV-2-specific CD8 T cells with a tissue-resident phenotype up to 2 months following viral clearance [40]. The identification of SARS-CoV-2 isolates bearing nonsynonymous mutations of MHC-I restricted-epitopes, likely driving selection of escape mutants, gave further support to the possibility of a protective role of CD8 T cell response [41], although this has to be viewed in the context of MHC polymorphism in the human population.

Nonetheless, increased SARS-CoV-2 antigen-specific CD8 T cells in the peripheral blood have also been associated with viral persistence after clinical recovery [42]. These puzzling observations might suggest that CD8 T cells did not play a role in clearing infection, although it might also be possible that increased CD8 T cell frequencies in these patients reflected repeated stimulation by particularly high viral loads and/or persistent viral RNA shedding [42], thus echoing the correlation found between disease severity and high serum titres of antibodies specific for the receptor binding domain (RBD) of spike [43]. Additionally, in contrast to the evidence in favour of a protective antigen-specific CD8 T cell function against SARS-CoV-2 are data showing that patients who developed post-COVID inflammatory lung sequalae had much higher frequencies of SARS-CoV-2-specific CD4 and CD8 T cells, and elevated plasma levels of IL-6 and C-reactive protein (CRP), when compared to those who completely recovered [44]. This supports the view that SARS-CoV-2-specific CD4 and CD8 T cells can contribute substantially to immunopathology (Figure 1A, in blue). Moreover, COVID-19 patients displaying a range of disease severities commonly showed profound alterations of peripheral blood naïve and memory T cell subsets, with lymphopenia concurrent with excessively high activation and proliferation [45]. These data might be seen to question the efficacy of T cell responses in the real world, “heat of the battle” against SARS-CoV-2, although they do not diminish the memory T cells’ potential importance upon pathogen encounter post-vaccination (Figure 1B, box 4, in bold), or post-infection (Figure 1A, box 4, in bold) [46,47].

SARS-CoV-2 vaccine-induced CD8 T cells might be expected to differ from those arising from natural infection, e.g., in terms of migratory pathways, antigen specificity, TCR repertoire (Figure 1). Reasons for these expected differences include the use of an intramuscular route for vaccine administration (which fails to adequately mimic airborne infection typical of SARS-CoV-2), and the fact that most of the currently used SARS-CoV-2 vaccines contain only the spike antigen, although vaccines with additional antigens are under development (Figure 1B) [14,15,48]. Furthermore, not all vaccines equally induce the different arms of adaptive immunity. A comparison among SARS-CoV-2 vaccines showed that classical vaccines composed of proteins and adjuvant elicited mostly antibodies, B cells, and helper CD4 T cells, while mRNA-based and adenovirus-based vaccines were able also to effectively prime naïve CD8 T cells, as anticipated from previous studies on these vaccine platforms (Figure 1B) [49,50].

A general consideration regarding vaccine-induced immunity is that the type of effector mechanism that is most needed for protection against an infectious disease depends, among other things, on the portal of entry and on the target organ of its causative pathogen. In the case of COVID-19, immune-mediated blocking of viral particles by neutralizing antibodies that hamper virus interaction with ACE-2, i.e., the SARS-CoV-2 receptor, and accelerate virus clearance, can be highly effective in attenuating infection. Thus, it is not surprising that serum anti-spike neutralizing antibodies, which are a subset of total serum anti-spike antibodies, are currently widely accepted as a correlate of protection [15]. However, cytotoxic CD8 T cell-mediated killing of virus-infected cells and CD8 T cell derived cytokine-mediated protective mechanisms can become prominent if neutralizing antibodies fail to sufficiently inhibit infection [14]. This appears to be the case at early times after the first dose of an mRNA vaccine. In fact, CD8 T cells specific for the spike protein of SARS-CoV-2 were found in the peripheral blood starting from day 10–12 after priming by Pfizer-BioNTech BNT162b2 mRNA (BNT) vaccine, as evaluated by MHC-I–peptide multimer staining, intracellular cytokine production, CD107a degranulation assay and flow cytometric analysis [51]. By contrast, serum-neutralizing antibodies were barely detectable at these early time points post-priming, and yet effective protection was already achieved [7,51].

Furthermore, T cell immunity can become central in immunocompromised patients lacking B cells and antibodies. Thus, it is encouraging that anti-SARS-CoV-2 vaccination induced higher levels of spike-specific CD8 T cells in a patient with X-linked agammaglobulinemia compared to healthy controls [52]. Likewise, in multiple sclerosis patients treated with B cell-depleting anti-CD20 monoclonal antibodies (mAbs), T cell responses were mostly preserved after SARS-CoV-2-vaccination, as evaluated by antigen-specific IFN-γ production [53]. However, low T cell responses were documented in common variable immune deficiency (CVID) patients displaying hypo- or agammaglobulinemia in the context of a more general impairment of the adaptive immunity [53,54]. Consequently, it has been suggested that effective strategies should be promptly implemented in these patients, e.g., by additional vaccine doses or treatment with anti-SARS-CoV-2 monoclonal antibodies in high-risk settings [54].

## 4. CD8 T Cell Responses to Prime/Boost Vaccination

Persistence of protective memory is generally achieved by administration of two or more vaccine doses, unless the vaccine is highly potent and able to provide life-long immunity with a single dose [55]. An example of a single-dose vaccine is the yellow fever vaccine (YFV-17D), which consists of a live attenuated pathogen. It has been shown that the magnitude of the CD8 T cell response positively correlated with increasing YFV-17D-attenuated viral load, up to a response plateau [56], and that this vaccine elicited long-lived CD8 T cells with open chromatin imprinting at effector gene loci [57]. Such durable memory stands in contrast to the one induced by vaccines that carry either low antigen amounts (e.g., protein and adjuvant), or antigen-encoding genetic material with limited in vivo expression (non-replicating viruses, short-lived mRNA). For vaccines of these types, a single injection (prime) is usually insufficient to obtain long-lived immunity, necessitating that most regimens be based on at least two injections several weeks or months apart (prime/boost vaccination) [58] (Figure 2A; note the difference between primary and secondary memory). Prime/boost regimen is termed homologous if an identical vaccine is used for both priming and boosting, and heterologous if the boost is performed with a vaccine carrying an antigen identical to that used for priming but delivered via a different vector [58]. Most SARS-CoV-2 vaccines were initially used according to a homologous prime/boost regimen, while Janssen Ad26.COV2.S (Ad26) was authorized as a single dose vaccine, although a boost vaccination was subsequently recommended for this vaccine also.

The exact mechanism by which prime/boost vaccination establishes CD8 T cell memory is not understood. While boosting can certainly increase the magnitude of antigen-specific CD8 T cell responses [59], it might also have an impact on epitope immunodominance [58,60]. For example, it was found that CD8 T cells against HIV-1 envelope protein induced by heterologous prime/boost vaccination had an increased breadth (number of antigenic epitopes recognized) and depth (variant coverage within an epitope) than those induced by a homologous regimen [61,62]. The relevance of these issues in designing vaccination strategies should not be underestimated, as epitope recognition can become central to cross-reactive protection across similar viruses, for example influenza viruses A, B and C [63]; a number of coronaviruses [64,65]; and pathogen variants, including those of influenza A virus [66] and of SARS-CoV-2 [67].

Prolonged persistence of already primed CD8 T cells, and further differentiation of these cells into long-lived memory CD8 T cells are key impacts of boosting [59,68]. Building on a linear model of memory CD8 T cell differentiation that envisions that naïve cells differentiate into effectors, and some of those effectors then differentiate into memory cells [55], Matzinger hypothesized the existence of a key “check” step, in which effector CD8 T cells would be tested for their ability to kill antigen-positive target cells. This step would enable only those effector CD8 T cells which can correctly perform their function (having adequate signalling machinery, cytoskeleton organization, granule content, etc.) to further differentiate and enter into a long-lived memory state, while preventing the useless and energy-consuming accumulation of unfit CD8 T cells [69]. Putting prime/boost vaccination in this frame, APC killing by primed CD8 T cells at the time of boost would be instrumental to establish long-lived memory, and to select the best-fit killers in the memory pool. Likewise, another “check” step may be exerted by regulatory T cells that may narrow the repertoire of memory T cell clones to those with very-high affinity and/or avidity, which may thereby perform better upon pathogen challenge [70]. Thus, vaccine design should be cognizant of its potential to influence the relative balance of regulatory and effector T cells.

Optimizing vector combinations and the order of injections may enhance the capacity of either homologous or heterologous regimens to induce durable CD8 T cell responses [71]. There are examples of heterologous prime/boost vaccinations out-performing homologous ones in experimental models [72,73] and in human studies. For example, it was shown that about 10 weeks after priming with AstraZeneca’s ChAdOx1-nCov-19 (ChAd) recombinant vector, boosting with BNT mRNA vaccine was more effective in stimulating CD8 T cells than boosting with ChAd [74]. In this setting, heterologous boosting resulted in higher spike-specific CD8 T cell frequencies, and increased percentages of IFN-γ^+^ CD8 T cells in peripheral blood at 2 weeks post-boost, while TNF-α^+^ CD8 T cells were comparably induced by both types of prime/boost vaccination [74]. One possible explanation for the reduced efficacy of homologous boost is that antibodies against the priming vector can accelerate vaccine clearance, thereby time-limiting T cell secondary exposure to antigens [58]. Therefore, it may be appropriate to apply a delayed boost regimen in homologous settings, thus taking advantage of antibody decline after priming (see also below).

Nonetheless, one important difference between naïve and memory T cells is that the former type of cells requires mature DCs to be primed, while the latter has less stringent requirements for activation, and may be boosted by B cells and/or other lymphocytes, including T cells or NK cells [75,76]. In the case of B cells, the presence of vector- and/or antigen-specific memory lymphocytes acting as potent APCs in the homologous boost setting may greatly increase T cell stimulation. Thus, it might not be surprising that there are several examples of effective performance by homologous prime/boost strategies [77], including many of the current COVID-19 vaccines [5,6,7]. Nonetheless, T cell responses to SARS-CoV-2 vaccination were mostly preserved in multiple sclerosis patients treated with anti-CD20 mAbs, suggesting that B cells were redundant APCs in these settings, perhaps owing to compensatory mechanisms. It is also possible that there had been sufficient B cell repopulation by the time that the vaccination occurred, i.e., from 3 to 6 months after anti-CD20 therapy [53].

Finally, prime/boost vaccinations via appropriate routes may favourably impact on the migratory pathways of primed CD8 T cells. This possibility can be exploited for the recruitment of memory CD8 T cells at mucosal sites, for example by combining subcutaneous priming with intranasal boosting [78], or intramuscular priming and intravaginal boosting [79], thus establishing abundant tissue-resident memory cells at the most appropriate anatomical sites. Another yet distinct type of prime/boost vaccination, named “prime-and-trap”, has proven effective in generating liver-resident memory CD8 T cells and protecting mice from malaria at the pre-erythrocytic stage [80]. In this experimental model, priming was mediated by a DNA gene-gun applied to abdominal skin, which induced recirculating CD8 T cells. Those T cells were then recruited to the liver following an intravenous boost with radiation-attenuated sporozoites, that are naturally liver-homing [80]. Vaccination regimens that specifically promote immunity in the target tissues of infection by intranasal administration are among the emerging COVID-19 vaccination strategies [81]. For example, a homologous combination of intramuscular priming and intranasal boosting with the Ad5-nCoV-S CanSinoBIO vaccine has just been approved in China, while another inhaled vaccine has been approved as a two-dose primary inoculation in India [81] (see also below).

## 5. Prime/Boost Vaccination during COVID-19 Pandemic: Strengths and Limitations

Vaccines against SARS-CoV-2 have been strikingly successful in preventing severe disease, hospitalization, and death [14,15]. Still, breakthrough infections have proven common some months after priming, or after completion of prime/boost vaccination [14,15]. This might not be surprising, considering that the United States Advisory Committee for Immunization Practices recommends administering more than two vaccine doses for the prevention of many infectious diseases in humans, and in some cases to administer boosters repeatedly [82].

In a longitudinal study to follow immune response kinetics following four different SARS-CoV-2 vaccines, it was found that neutralizing antibodies were extremely high at 1 month after completion of prime/boost vaccination with mRNA vaccines (BNT and Moderna mRNA-1273), and were still present at 6 months, albeit a decrease was evident; prime/boost vaccination with recombinant protein-based, adjuvanted Novavax NVX-CoV2373 (NVX) resulted in titres similar to those obtained with mRNA vaccines at 6 months; while those induced by priming with the adenovirus-based vaccine Ad26 were lower [50]. All four vaccines induced peripheral blood CD4 and CD8 T cell responses, as measured by activation-induced marker (AIM) expression and intracellular cytokine production after in vitro stimulation with spike peptide pools. However, Ad26 priming resulted in evidently lower CD4 responses, while CD8 T cell responses induced by Ad26 priming tended to remain stable or even to slightly increase over 6 months. As expected, prime/boost vaccination with protein-adjuvant-based NVX induced weak CD8 T cell responses [50]. Studies are ongoing to better understand the differential decay of distinct arms of vaccine-induced immunity, and the implications for the anti-SARS-CoV-2 protection of such unequal waning of immunological memory.

The common occurrence of breakthrough infections, concurrently with the continuous emergence of SARS-CoV-2 VOCs with enhanced transmission, has raised the concern that VOCs can escape immune recognition owing to a combination of quantitatively declining immunity and qualitative epitope mutation, particularly given that large-scale vaccination campaigns have been carried out with the vaccines that are based on the ancestral virus (Wuhan-like). Luckily, alpha, beta, gamma and delta VOCs only partially escaped neutralizing antibodies induced by SARS-CoV-2 vaccines in widespread use; it was demonstrated that the in vitro neutralizing activity against the ancestral virus was predictive of neutralization of these VOCs, and correlated with protection [83]. Furthermore, a recent study showed that the emergence of VOCs from alpha to omicron had a more prominent impact on antibody recognition and B cell response than on CD4 and CD8 T cell responses at 6 months after vaccination with four different SARS-CoV-2 vaccines [67], as expected considering that antibodies recognize a few epitopes on the virus’ surface, while CD4 and CD8 T cells a wide repertoire of viral epitopes presented by APCs or infected cells. For example, predicted preservation of SARS-CoV-2 spike T cell epitopes in omicron was 72% for CD4 T cells and 86% for CD8 T cells, and experimental assessment revealed an overall conservation of memory CD4 and CD8 T cell recognition of the omicron spike protein [67]. These results are in agreement with an additional study that focused on anti-omicron immunity [84]. It should be noted that viral variants able to evade T cell recognition are much less likely to be selected than those escaping antibody recognition, considering that HLA-I genes are extremely polymorphic. Furthermore, SARS-CoV-2 infection has an acute clinical course in the majority of cases, such that in any one individual with a given HLA haplotype, there is insufficient time for the generation and selection of viral mutants.

The ongoing COVID-19 pandemic has drawn our attention to an interesting phenomenon that deserves further investigation. Either occurrence of breakthrough COVID-19 infections in vaccinated individuals, or vaccination of previously infected persons, can result in a very strong synergy, leading to so-called “hybrid immunity” [85,86]. The mechanisms that make the combination of vaccine- and infection-generated immunity highly potent are under study and may include an increased breadth of B and T cell responses, and the strengthening of protection at the pathogen’s portal of entry, as natural infection results in abundant mucosal IgA, and CD4 and CD8 T_RM_ cells [85,87].

## 6. Additional Boosters and Long-Lived Memory CD8 T Cells

To prolong anti-COVID-19 protection against the ancestral virus and its VOCs, a third, and fourth, and in some countries even a fifth vaccine dose have been administered. Repeated vaccine administrations can also give the opportunity to enhance the breadth of the response, when appropriate combinations of vectors and inter-dose time intervals are used [88]. A side-by-side comparison was performed in fully vaccinated individuals who had received a prime/boost vaccination with one of three COVID-19 vaccines (i.e., BNT, mRNA-1273, ChAd) and at least 12 weeks later received either a homologous or a heterologous (third dose) booster. Experiments performed with pre-booster peripheral blood mononuclear cell (PBMC) samples collected on the day of the third dose showed that antigen-specific CD8 T cells producing IFN-γ and/or IL-2 were induced more effectively after prime/boost vaccination with ChAd than after prime/boost vaccination with either BNT or mRNA-1273 vaccines [89]. Experiments performed at 15 days post-booster showed that the lower CD8 T cell response induced by prime/boost vaccination with BNT was boosted in both homologous and heterologous settings, with similar results obtained with the mRNA-1273 vaccine [89]. Notably, the homologous third dose booster did not further increase the ChAd prime/boost-elicited CD8 T cell response, whereas the heterologous one with either BNT or mRNA-1273 did increase it [89], thus echoing the above-mentioned study on ChAd/BNT prime/boost vaccination [74]. Furthermore, the lack of increased T cell immunity when the ChAd homologous booster was used after ChAd/ChAd prime/boost vaccination was consistent with results from another study [90]. 

The possibility that intranasal boosters can improve local protection at the pathogen portal of entry is under investigation by different research teams. Encouragingly, protection against SARS-CoV-2 was obtained and lung CD8 T_RM_ cells were generated in experimental models of vaccination after administering an intranasal booster to mice, previously primed and boosted with an mRNA-based vaccine via the intramuscular route [91]. Research is ongoing in relation to vaccination against SARS-CoV-2 via the nasal route, which is expected to protect the host at the pathogen portal of entry, thereby limiting lung pathology, and also reducing viral shedding from the upper respiratory tract, thereby decreasing inter-individual virus spread [81,92].

## 7. Time Interval between Vaccine Doses and CD8 T Cell Differentiation

The recommended intervals between doses of different vaccines for children and adults span a wide range, from only a few weeks to several months, or even years in some instances of booster doses (see current vaccine schedules recommended by the United States Centers for Disease Control and Prevention at https://www.cdc.gov/vaccines/schedules/index.html, accessed on 30 September 2022). Unfortunately, there are only scarce data sets regarding the impact of inter-dose time interval on T cell responses, as most immunogenicity studies of human vaccines focused only on antibodies. Furthermore, studies were not usually designed to specifically address the question of inter-dose interval. Rodrigues and Plotkin reviewed several trials of vaccination against human Papilloma virus, varicella-zoster virus, polio virus, and pneumococcus, and compared available data on antibody titres which were obtained using different inter-dose intervals [93]. The consensus emerging from this analysis was that a delayed prime/boost interval resulted in higher antibody titres, at least when a booster was administered within 6 months [93]. A better performance of delayed booster for antibody response was also predicted by computational models of immune response to vaccination [94].

During the COVID-19 pandemic, the impact of either short (3–4 weeks) or long (10–12 weeks) prime/boost intervals on antibody titres and T cell immunity became a hot topic. Taking into consideration vaccine shortages, delaying the boosts to about 12 weeks post-prime was a public health measure used in some countries to provide at least one vaccine dose to the highest possible number of individuals in the shortest possible timeframe. However, concerns were raised about the possibility of insufficient immediate protection of primed-only individuals, and/or of inadequate long-term immunity of those vaccinated according to a delayed boost regimen. We shall consider here the impact of prime/boost intervals on memory CD8 T cell differentiation by discussing firstly experimental findings in animal models, and secondly recent analyses in SARS-CoV-2 vaccinated individuals.

With regard to CD8 T cell differentiation, the time interval between repeated stimulations has been investigated in numerous experimental models over the years [95,96,97]. In some studies results were obtained in mice infected intravenously with replicating microbes in heterologous settings. For example, in a model using recombinant *Listeria monocytogenes*, *Vesicular stomatitis* virus, and *Vaccinia* virus, each encoding ovalbumin (OVA) as a shared antigen, it was shown that repeated exposures to OVA carried by different vectors resulted in reduced memory CD8 T cell longevity when the prime/boost interval was short, i.e., 2 weeks [97]. In other studies, mice were vaccinated subcutaneously with non-replicating vectors carrying OVA, and it was shown that one homologous boost was not effective when performed before the end of the acute primary response, i.e., at week 1 post-prime, whereas it enhanced CD8 T cell responses when performed at a later time-point, i.e., at week 3, week 6, or week 9 post-prime, with no differences between those three time points [96]. Emerging concepts in the field include the possibility that repeatedly stimulated CD8 T cells can rapidly lose their replicative potential if there is insufficient recovery time between stimulations. This may relate to the potential for chronically activated T cells to enter an exhausted state, as considered above. Additionally, the potency of recall responses seems to depend on the phenotype of the memory CD8 T cell population induced by priming, e.g., its epigenetic, transcriptional and/or metabolic state(s), which will vary at different times after priming [55,97,98,99].

We have investigated the impact of two inter-dose time intervals, both ≥1 month, on antigen-specific CD8 T cell response using a mouse model of heterologous prime/boost vaccination against HIV-1 gag. Specifically, there were two intramuscular injections with gag-encoding replication-defective vectors: Chimpanzee adeno-vector for priming, and modified vaccinia virus Ankara for boosting [100]. In this model, we compared boosting at day 30 with that at day 100 post-prime and found that the latter resulted in a higher frequency and functional capacity of gag-specific CD8 T cells, as evaluated at day 45 post-boost [100]. We found that the increased responsiveness of primed CD8 T cells to boosting at day 100 was associated with a distinct molecular signature, which trended towards a T_CM_ phenotype, and was characterized by a complete shut-off of a large set of mitotic genes, down-regulation of effector genes, and increase in genes regulating quiescence and metabolism. The up-regulation of quiescent genes might be particularly important to preserve the cells’ proliferative and metabolic potential, at the same time as maintaining the cells in a highly responsive state [101]. Furthermore, at day 100 there was a spatial redistribution of gag-specific CD8 T cells, so that their numbers decreased in blood and spleen, and increased in LNs and BM [100]. Altogether, our results documented a time-dependent maturation of primed CD8 T cells long after the end of the acute response to priming [100,102], coinciding with an active regulation of cell localization in peripheral lymphoid organs and in the BM, in agreement with a prominent contribution of BM to T cell memory [101]. Notably, our results support the notion that a central memory/ stem cell memory CD8 T cell phenotype is acquired slowly after vaccination [55], echoing recent findings obtained in COVID-19-vaccinated individuals [103].

A few recent COVID-19 vaccine studies addressed the effect of different inter-dose intervals on T cell immunity, in parallel with antibody responses. These are mostly post hoc exploratory analyses, owing to the practical reality of delayed boosting under the pressure of the pandemic emergency and vaccine shortages [93]. In some instances, primary and secondary memory were compared for individuals vaccinated according to a delayed boost regimen, aiming at quantifying the decrease in T cell immunity of primed-only individuals, as compared to those primed and boosted (Figure 2B). It was found that prior to boosting, many individuals had largely preserved their T cell responses that were further augmented following the second vaccine dose, whereas antibody responses waned rapidly after priming but were strikingly increased after boosting [104].

Alternatively, in most cases a comparison between short and long prime/boost intervals was made at the same time after boosting, in order to identify whether secondary memory T cell response kinetics and magnitude were different following the two regimens of prime/boost vaccination (Figure 2C) [104,105,106]. In general, it was found that peripheral blood T cell responses were robust, with some differences between the two vaccination regimens, and an interesting dichotomy when comparing T cell and antibody responses. More specifically, an insightful randomized immunogenicity trial comparing a short interval of 4 weeks with a longer one of 12 weeks in an adult cohort of individuals vaccinated with four different vaccine combinations (ChAd/ChAd, BNT/BNT, ChAd/BNT and BNT/ChAd) showed that with each combination T cell responses were higher in the short interval group, as evaluated by IFN-γ^+^ ELISPOT at 1 month post-boost, whereas antibody responses were higher after delayed boosting [105] (Table 1). These differences were mostly maintained at 6 months post-boost, despite some memory decay. The strongest T cell response was obtained with the ChAd/BNT immunization with the short-boost regimen, and the strongest antibody response with BNT/BNT with the long-boost regimen (Table 1). CD8 T cells were not specifically investigated in this study [105].

In one study investigating CD8 and CD4 T cells separately in the BNT/BNT setting, a short interval of 2–5 weeks more effectively induced IFN-γ^+^ CD8 T cells than a longer interval of 6–14 weeks, as evaluated by intracellular cytokine staining at 4 weeks post-boost [104]. Curiously, the reverse was observed for CD4 T cells producing IL-2, IFN-γ, or TNF-α [104]. Another study in a cohort of individuals of ≥80 years reported that BNT/BNT vaccination induced a higher T cell response following a short inter-dose interval of 3 weeks compared to a longer one of 11–12 weeks, evaluated by IFN-γ^+^ ELISPOT at 2–3 weeks post-boost [106]. However, at 10–11 weeks post-boost, this high response obtained by boosting after a short inter-dose interval significantly declined; unfortunately, the group vaccinated according to a long inter-dose interval regimen was not similarly tested at a later time point to check for memory persistence [106]. Further studies are certainly required for an in-depth assessment of CD4 and CD8 T cell immunity by multiple assays at several time points, possibly including longer prime/boost intervals, and varying schedules of repeated boosts.

## 8. Conclusions

In this review, we discussed the evidence for a protective role of antigen-specific CD8 T cells against SARS-CoV-2 infection, and how best to elicit such cells in appropriate quantity, quality and anatomical localization by vaccination, highlighting open questions in the field (Appendix A). It should be emphasized that CD8 T cells represent only one component of a multi-layered immune response. The Swiss cheese model originally conceptualized by the psychologist James Reason to explain complex systems such as nuclear power, and then applied by the virologist Ian Mackay to anti-COVID-19 interventions in public health, can also be perfectly applied to the complexity and redundancy of adaptive immune mechanisms against SARS-CoV-2, as recently illustrated by an illuminating scheme on correlates of protection [15]. According to this scheme, neutralizing antibodies, CD4 T cells, circulating CD8 T cells, CD8 T_RM_ cells, etc., each provide a layer of protection, so that even though there might be a hole in each layer, it is unlikely that those holes are lined up and that the virus can thereby evade all forms of adaptive immunity [15].

In sum, the contribution of CD8 T cells to anti-SARS-CoV-2 immunological memory of different types, i.e., after infection, vaccination, or a combination of both, is now emerging as an important layer of protection, particularly in the following circumstances: (i) early phase after priming, before increases in antibody titres; (ii) long after infection or vaccination, when neutralizing antibodies wane; (iii) in individuals with compromised B cell and antibody responses, due to immunodeficiency or therapy; (iv) for protection against VOCs that have been selected for their evasion of neutralizing antibodies; and (v) for prevention of inter-individual transmission and protection against virus-induced pathology at the pathogen’s portal of entry. In respect to vaccination, it is clear that multiple vaccine doses promote differentiation of memory CD8 T cells, although CD8 T cell memory magnitude and longevity may be substantially increased only when the inter-dose time interval is adequate. Large-scale T cell response monitoring will be instrumental for improving vaccination strategies against SARS-CoV-2. In this context, it is encouraging that quantitative assessment of T cell immunity is technically improving, e.g., sets of antigenic peptides have been validated, and novel rapid assays are available [107,108,109] Finally, it is of note that CD4 and CD8 T cells induced by a chimeric spike–nucleoprotein vaccine were able to protect against SARS-CoV-2 infection independently of neutralizing antibodies in mice and hamster experimental models, paving the way to new strategies to improve human vaccines against COVID-19 by strongly promoting T cell responses in addition to B cells and antibodies [110].

## Figures and Tables

**Figure 1 ijms-23-14367-f001:**
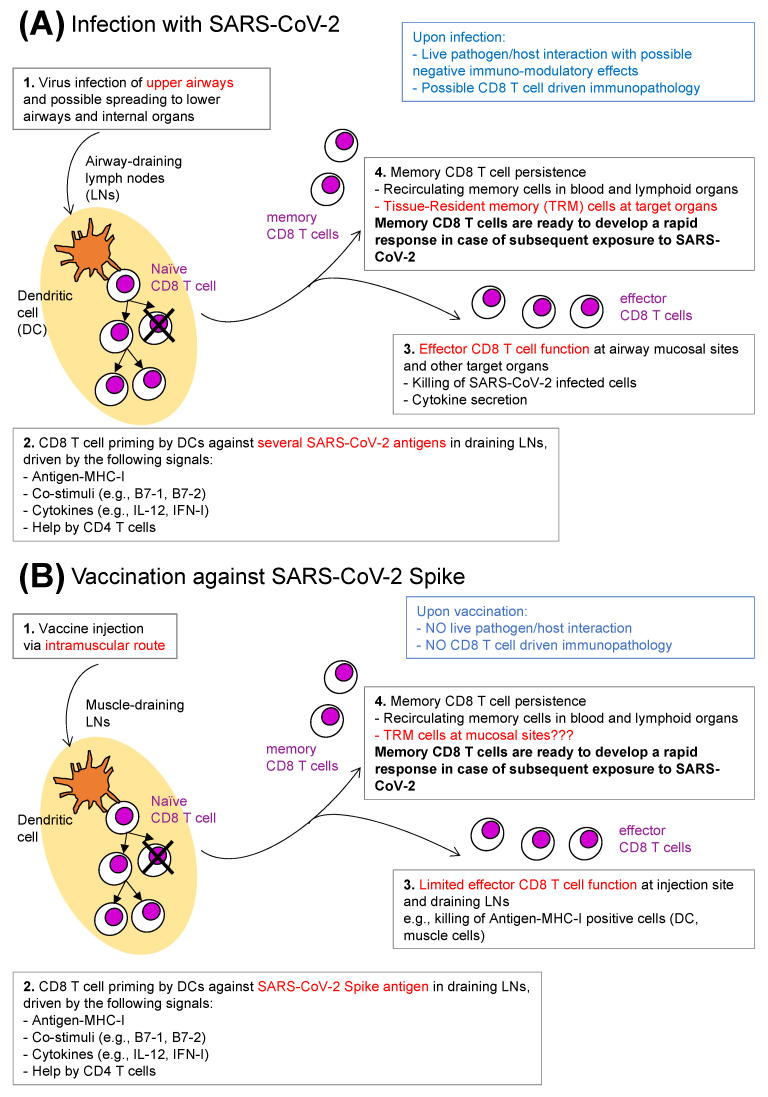
**Anti-SARS-CoV-2 CD8 T cells: natural infection versus vaccination.** (**A**) **Infection with SARS-CoV-2.** SARS-CoV-2 infection usually starts in the upper respiratory airways, with possible subsequent virus spreading to other organs (**1**). Priming of naïve CD8 T cells occurs in airway-draining lymph nodes (LNs), wherein mature dendritic cells (DCs) present epitopes from a series of SARS-CoV-2 antigens in the context of MHC-I molecules, together with ligands for costimulatory receptors, and in the presence of additional signals provided by helper CD4 T cells and cytokines (**2**). Primed CD8 T cells proliferate and differentiate, generating a progeny of CD8 T cells that migrate out of LNs and reach distant sites all over the body. Upon antigen–MHC-I recognition on the surface of SARS-CoV-2-infected cells in so-called target organs, short-lived effector CD8 T cells exert their function, e.g., cytotoxicity and cytokine secretion (**3**). Memory CD8 T cells recirculate in blood and lymphoid organs and are found as differentiated tissue-resident memory CD8 T cells in target organs (**4**). Possible limitations of the CD8 T cell-mediated protective function against SARS-CoV-2 infection are listed in blue. (**B**) **Vaccination against SARS-CoV-2 spike.** A typical CD8 T cell response elicited by a vaccine encoding for the spike antigen of SARS-CoV-2 (e.g., adenovirus- or mRNA-based vaccine) is schematically depicted. After vaccine injection via the intramuscular route (**1**), the few naïve CD8 T cells that are specific for spike epitopes clonally expand and differentiate in muscle-draining LNs, wherein CD8 T cell priming occurs (**2**). Effector CD8 T cell function is expected to be very limited after a single vaccine injection, and more pronounced after repeated vaccine doses; it mostly consists of killing of spike-expressing muscle cells and DCs, and of DCs that expose spike-derived peptides in the context of MHC-I after taking up debris from damaged spike-expressing cells (**3**). Anti-spike vaccines elicit memory CD8 T cells recirculating in blood and lymphoid organs, and possibly tissue-resident memory CD8 T cells at mucosal sites (**4**). ((**A**) vs. (**B**)). Memory CD8 T cells which persist either post-infection (**A**) or post-vaccination (**B**) develop a rapid response in case of subsequent encounter with SARS-CoV-2, contributing to protection (box 4 in (**A**,**B**), in bold). The main differences between A and B are highlighted in red (boxes **1**–**4** in (**A**,**B**)). Other differences remain to be determined, e.g., in respect to the quality/quantity of the priming signals listed in box 2 in A and B, to the durability of memory CD8 T cell response, etc. (not depicted). The limitations of CD8 T cell protective function listed in (**A**) are not applicable in (**B**) (in blue). See text for more details.

**Figure 2 ijms-23-14367-f002:**
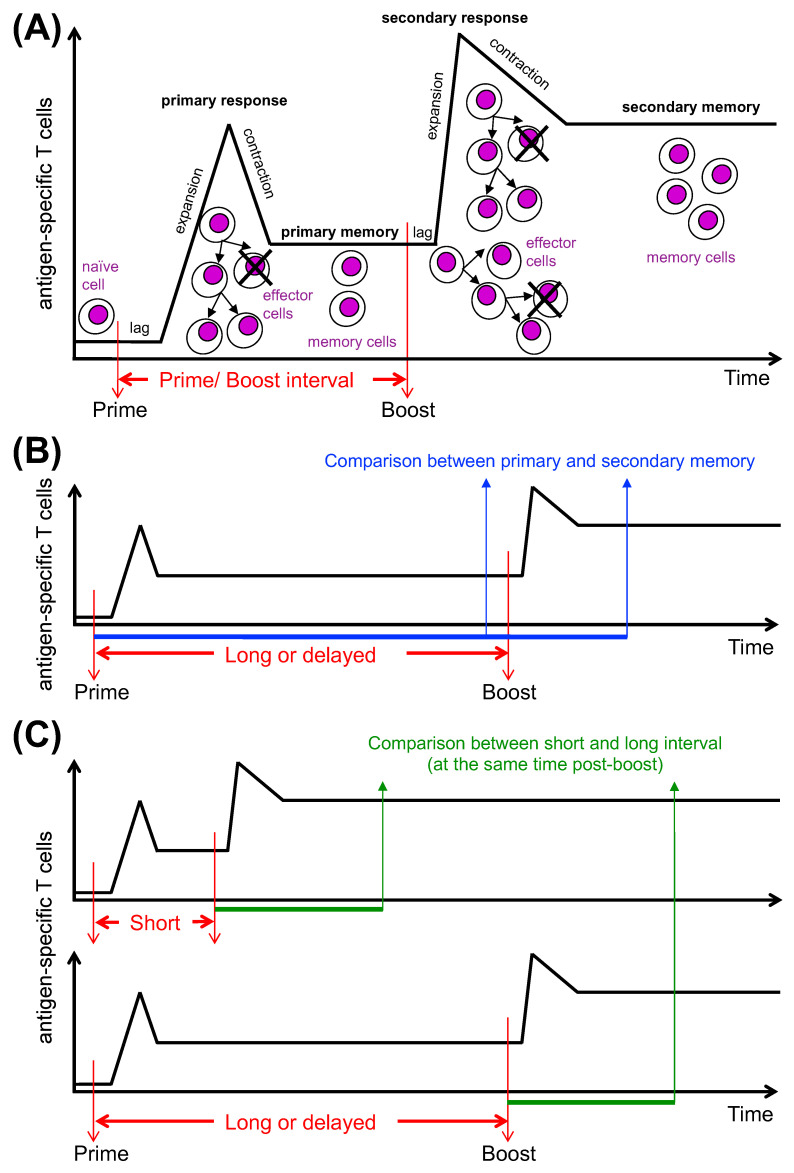
**Antigen-specific T cell response by prime/boost vaccination**. (**A**) **Kinetics of response**. After vaccine priming, the few naïve T cells that are specific for vaccine-carried antigens clonally expand and differentiate. The number of antigen-specific T cells rises gradually to detectable levels after a time lag, reaching a peak in some days. At the end of the acute phase of the primary response, most effector T cells die during the 
contraction phase, leaving behind a few memory T cells that remain in the primary memory phase. Upon vaccine boosting, these cells start proliferating after a shorter time lag and reach a higher peak compared with naïve cells. The contraction phase leaves behind a high number of memory T cells that persist in the secondary memory phase. (**B**,**C**) **Prime/boost intervals of different time lengths: insightful comparisons.** The panels represent T cell response to prime/boost vaccination according to either a “long” (also called “delayed”) or a “short” inter-dose interval regimen. Please note that, although it is possible that the two regimens result in different T cell responses, no changes in antigen-specific T cell numbers are depicted for simplicity. (**B**). Comparison between primary (pre-boost) and secondary (post-boost) memory T cell 
responses in the vaccination setting with a long interval. (**C**). Comparison between secondary memory T cell response after vaccination according to a short interval regimen and secondary memory T cell response after vaccination according to a long interval regimen (as evaluated at the same time 
post-boost).

**Table 1 ijms-23-14367-t001:** T cell and antibody response following anti-SARS-CoV-2 vaccination performed with four different vaccine combinations and two time-lengths of prime/boost interval. This table is based on a publication selected because at this time, to our knowledge, it is unique in making the comparison of four vaccine combinations and two time intervals in a randomized study [105]. To assess the effects of short versus long prime/boost interval on vaccinations performed with different vaccine combinations, adult volunteers were randomly divided into eight groups. Each group was vaccinated with one of four different prime/boost vaccine combinations (ChAd/ChAd, BNT/BNT, ChAd/BNT and BNT/ChAd), each performed according to either a 4-week or a 12-week prime/boost interval. Serum antibody titres were measured, and T cell responses assayed by IFN-γ ELISPOT at different times after boosting, up to 7 months. The table is based on quantitative data at 1-month post-boost. A slow decline was observed at later time points, nevertheless, differences between the short and long interval for each vaccine combination were maintained overall. For more details see original publication [105].

	T Cell IFN-γ ELISPOT		Anti-Spike IgG	
	Short Interval	Long Interval	Short Interval	Long Interval
ChAd/ChAd	++	+	+	++
BNT/BNT	+++	++	++++	+++++
ChAd/BNT	+++++	++++	++++	++++
BNT/ChAd	++++	+	+++	++++

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
