# Peer review of "Durable CD8 T Cell Memory against SARS-CoV-2 by Prime/Boost and Multi-Dose Vaccination: Considerations on Inter-Dose Time Intervals"

_ijms, 2022, doi:10.3390/ijms232214367_

Round 1

Reviewer 1 Report

The review presented by Ambra Natalini et al. makes a positive impression and will be useful to researchers developing vaccines not only against SARS-CoV-2, but also other pathogens.

Despite the extensive information presented in the review, I would like to note the extremely scarce illustrative material: 1 figure and 1 table. Verbal descriptions of the interaction of immune cells with pathogens and among themselves are difficult to understand for a non-core specialist. The overview would be much better with the addition of illustrative schemes of the described immune systems and signal sequences.

Author Response

We thank Reviewer 1 for her/his positive feedback.

We have addressed Reviewer 1 comment by adding a new figure representing an illustrative scheme of anti-SARS-CoV-2 CD8 T cell response after natural infection, or vaccination.

Reviewer 2 Report

This is an excellent review on the CD8 T cell responses against SARS-CoV-2 by prime/boost and multi-dose vaccine. The authors start with the concise introduction of the fundamentals of T cell responses and then summarise the recent findings in the literature. Furthermore, the authors offer their own insightful discussion on these findings. The review could be very useful not only for the experts but also for any new starting Master and PhD students in the field.

Author Response

We thank Reviewer 2 for her/his positive feedback.

Round 2

Reviewer 1 Report

The authors took into account my wishes. This manuscript may now be published in the form presented.